# Transposable element insertions in 1000 Swedish individuals

**Kristine Bilgrav Saether**[1,2], **Daniel Nilsson**[1,2,3], **Håkan Thonberg**[1,3], **Emma Tham**[1,3], **Adam Ameur**[4], **Jesper Eisfeldt**[1,2,3]*, **Anna Lindstrand**[1,3]

**1** Department of Molecular Medicine and Surgery, Karolinska Institutet, Stockholm, Sweden, **2** Science for Life Laboratory, Department of Molecular Medicine and Surgery, Karolinska Institutet, Stockholm, Sweden, **3** Department of Clinical Genetics, Karolinska University Hospital, Stockholm, Sweden, **4** Science for Life Laboratory, Department of Immunology, Genetics and Pathology, Uppsala University, Uppsala, Sweden

* jesper.eisfeldt@scilifelab.se

**Data Availability Statement:** The SweGen transposable element (TE) database is available at the SweFreq project website https://swefreq.nbis.se/. The 1000 Genomes Project TE database is available at Zenodo, via doi: 10.5281/zenodo.

## Abstract

The majority of rare diseases are genetic, and regardless of advanced high-throughput genomics-based investigations, 60% of patients remain undiagnosed. A major factor limiting our ability to identify disease-causing alterations is a poor understanding of the morbid and normal human genome. A major genomic contributor of which function and distribution remain largely unstudied are the transposable elements (TE), which constitute 50% of our genome. Here we aim to resolve this knowledge gap and increase the diagnostic yield of rare disease patients investigated with clinical genome sequencing. To this end we characterized TE insertions in 1000 Swedish individuals from the SweGen dataset and 2504 individuals from the 1000 Genomes Project (1KGP), creating seven population-specific TE insertion databases. Of note, 66% of TE insertions in SweGen were present at >1% in the 1KGP databases, proving that most insertions are common across populations. Focusing on the rare TE insertions, we show that even though ~0.7% of those insertions affect protein coding genes, they rarely affect known disease casing genes (<0.1%). Finally, we applied a TE insertion identification workflow on two clinical cases where disease causing TE insertions were suspected and could verify the presence of pathogenic TE insertions in both. Altogether we demonstrate the importance of TE insertion detection and highlight possible clinical implications in rare disease diagnostics.

## Introduction

In recent years, high-throughput genomics-based approaches such as whole genome sequencing (WGS) has been a great success with overall diagnostic yields of between 20 and 55% for mixed rare diseases patient groups [1]. Although highly successful, an average of 60% of rare disease patients investigated with clinical WGS still do not receive a molecular diagnosis [2]. The reasons for this could be due to the specific disease-causing gene is not described or that the underlying cause is not (mono)genetic. However, equally important is a lack of knowledge regarding the human genome composition and structure [3], such as noncoding variants, as

7875363. All raw WGS data is publicly available. The SweGen dataset is available at https://swefreq.nbis.se/ upon signing a data access agreement. The 1000 Genomes Project data is openly available at https://www.internationalgenome.org/.

**Funding:** The work was supported by the Swedish Research Council (2017-02936, 2019- 395 02079), Karolinska Institutet and the Stockholm Region (FoUI-961630, FoUI-954569). The funders had no role in study design, data collection and analysis, decision to publish, or preparation of the manuscript.

**Competing interests:** The authors have declared that no competing interests exist.

well as epigenetic modifications [4, 5]. To correctly interpret individual phenotype-genotype interactions we first need to study the normal genome architecture.

The human genome contains an abundance of genetic variation, including single nucleotide variants (SNVs), small insertions and deletions (INDELS) as well as structural variation (SV). SVs, generally defined as genetic variations larger than 50 base-pair (bp), contribute the most to the genetic difference between individuals, as each SV comprise many bases [6]. There are multiple SV subtypes such as copy number variants, insertions, inversion, translocations [7], and one important, but understudied SV subgroup are the transposable element (TE) insertions. TE are DNA sequences that compose around 50% of the human genome [8].

TE are DNA sequences that can change their position in the genome. Two main classes are described, DNA transposons (Class II TE), which move through a cut and paste mechanism, make up less than 2% of human genomes, and the more common retrotransposons (RTs, Class I TE) that move through RNA intermediates. These are further subdivided into long terminal repeat (LTR) or non-LTR retrotransposons, characterized by the presence or absence of LTRs [9].

The main LTR-RT in humans is the 9.5 kb human endogenous retrovirus (HERV), that contains the four genes *gag*, *pro*, *pol* and *env* flanked by LTRs. HERVs are mostly found in heterochromatin and are often silenced by epigenetic signaling. Even so, HERV activity and transcription has been described in human auto-immunity disorders as well as in cancer [3]. The three main non-LTR RTs in humans are (I) the 6 kb long-interspersed nuclear element 1 (L1), (II) the 300 bp short-interspersed elements, SINEs (*Alu*) and (III) the 2 kb SINE-VNTR-*Alu* (SVA). Of these, only the L1 elements are autonomous with functional transposon activity [10]. They encode an RNA II promoter and encode proteins with both an endonuclease and reverse transcriptase enabling reverse transcription into cDNA and integration into the genome, a process called target primed reverse transcription (TPRT). The L1 proteins are cis-preferent and prefer to transpose the RNA that encoded it over RNA from other L1 sequences. In contrast, *Alu* elements, making up 11% of the human genome [11], and SVAs are non-autonomous and utilize the L1 machinery to enable presence at new genomic locations [10], a process called in *trans* retrotransposition. The germline *de novo* rate of TE insertions is not entirely determined, recent estimate range from 1:29–40 births for *Alu*, 1:63–117 births for L1 and 1:63–206 births for SVA [12, 13]. This is low compared to SNVs where the average *de novo* mutation rate has been estimated to 73 *de novo* SNVs per birth (of which 2–4 are coding) [14].

TE insertions can disrupt genes, cause frameshift mutations and alter splicing patterns, and depending on the location in the genome may cause monogenetic disease [15–17]. An example is Duchenne Muscular Dystrophy (*DMD*, MIM# 310200), where several instances of (3') L1 element insertion cause exon skipping or -shuffling, resulting in disease [10, 18–20]. In another example, an SVA element was found to cause exon-trapping in *MFSD8* (or *CLN7* MIM# 610951) in a child with Batten disease, a defect resolved by a personalized antisense-oligonucleotide drug [21].

TE insertions have been characterized in short read WGS by the 1000 Genomes Consortium (1KGP) and in the Simons Genome Diversity Project (SGDP) [6, 22, 23]. Despite this resource and the known potential for TE insertions to cause disease, none have yet implemented TE insertion calling in clinical use. With an aim to increase the diagnostic yield, we first characterize TE insertions in two population cohorts sequenced with short-read WGS. Specifically we analyzed 1000 Swedish individuals in the SweGen dataset [24] and 2504 mixed-population individuals from the 1000 Genomes Project (1KGP) [25], and constructed population-based TE insertion databases for both. These were proven useful when testing our TE

insertion calling and filtering workflow on two clinical cases, where we successfully identified disease-causing *Alu* insertions.

## Results

### Distribution of detected TE insertions

*Alu*, HERV, L1 and SVA elements absent from GRCh38 were detected in WGS from datasets SweGen (n = 1000) and 1KGP (n = 2405) using a specialized TE insertion caller, RetroSeq [26] (Fig 1).

Across the 1KGP dataset, a total of 105 742 TE insertions, of which 84 614 *Alu*, 698 HERV, 18 367 L1 and 2071 SVA were found in more than 1% of individuals. This results in a per individual median of 4202 *Alu*, 108 HERV, 1661 L1 and 379 SVA insertions (Table 1, Fig 2A–2D). Next, in SweGen we detected a total of 135 719 TE insertions, of which 82 392 *Alu*, 997 HERV, 49 802 L1 and 1597 SVA present in more than 1% of individuals, with a sample median of 3460 *Alu*, 193 HERV, 1127 L1 and 195 SVA insertions (Fig 2E, Table 1).

TE insertions are believed to be enriched in certain genomic regions, such as in non-coding regions, due to selective forces [27]. The identified TE insertions were separated into 1 MB bins and plotted in a circos plot. The distribution across the genome was similar between the datasets (Fig 3, outer circles). High-density regions were found on chromosomes 1, 6 and 7 for both datasets, as well as 19 in 1KGP (Fig 3). The median insertions in a 1MB bin was 37 for SweGen and 24 for 1KGP (Fig 3). Removing rare insertions (insertions with population frequency <5%), resulted in a more even distribution across the genome and a decrease of high-density regions. The insertion bin count decreased with 86% for SweGen and with 71% for 1KGP (median bin count of 5 and 7 respectively) (Fig 3). Some high-density regions remained, correlating with chromosomal regions known to have high sequence diversity and gene density.

### Distribution of TE insertions and allele frequencies across populations

In rare disease diagnostics, frequency databases are often used to remove common variants. Frequency databases can be used to annotate patient variants with their frequency in the population. It is therefore important to have databases that capture the common variants in a population, for an efficient rare disease diagnostic analysis.

Separate databases were constructed for TE insertions detected in the SweGen dataset, the full 1KGP dataset, as well as one for each 1KGP subpopulation (AFR African, AMR American, EAS East Asian, EUR European, SAS South Asian) (Fig 1).

Using the six 1KGP databases, we characterized the 1KGP frequencies of the TE insertions identified in the SweGen dataset (Fig 4). For HERV and L1, 0.35–0.65 (35–65%) of TE insertions were unique to SweGen using the various 1KGP databases (Fig 4B and 4C). *Alu* and SVA insertions called across the SweGen data were mostly represented in the 1KGP databases, where <0.2 (20%) were regarded as novel (Fig 4A and 4D). When considering all TE insertions with a 1KGP database frequency >1% (0.01), >66% of TE insertions could be excluded.

### TE insertions in disease causing genes

In addition to using allele frequencies to narrow down potential disease-causing variants in a diagnostic setting, gene lists are commonly used for *in silico* filtering [2]. The individuals in the SweGen cohort represent the general Swedish population, and by applying clinically relevant gene lists on all SweGen TE insertions found in the 1KGP database, we assess the usefulness of gene lists for analysis of TE insertions in clinical diagnostics.

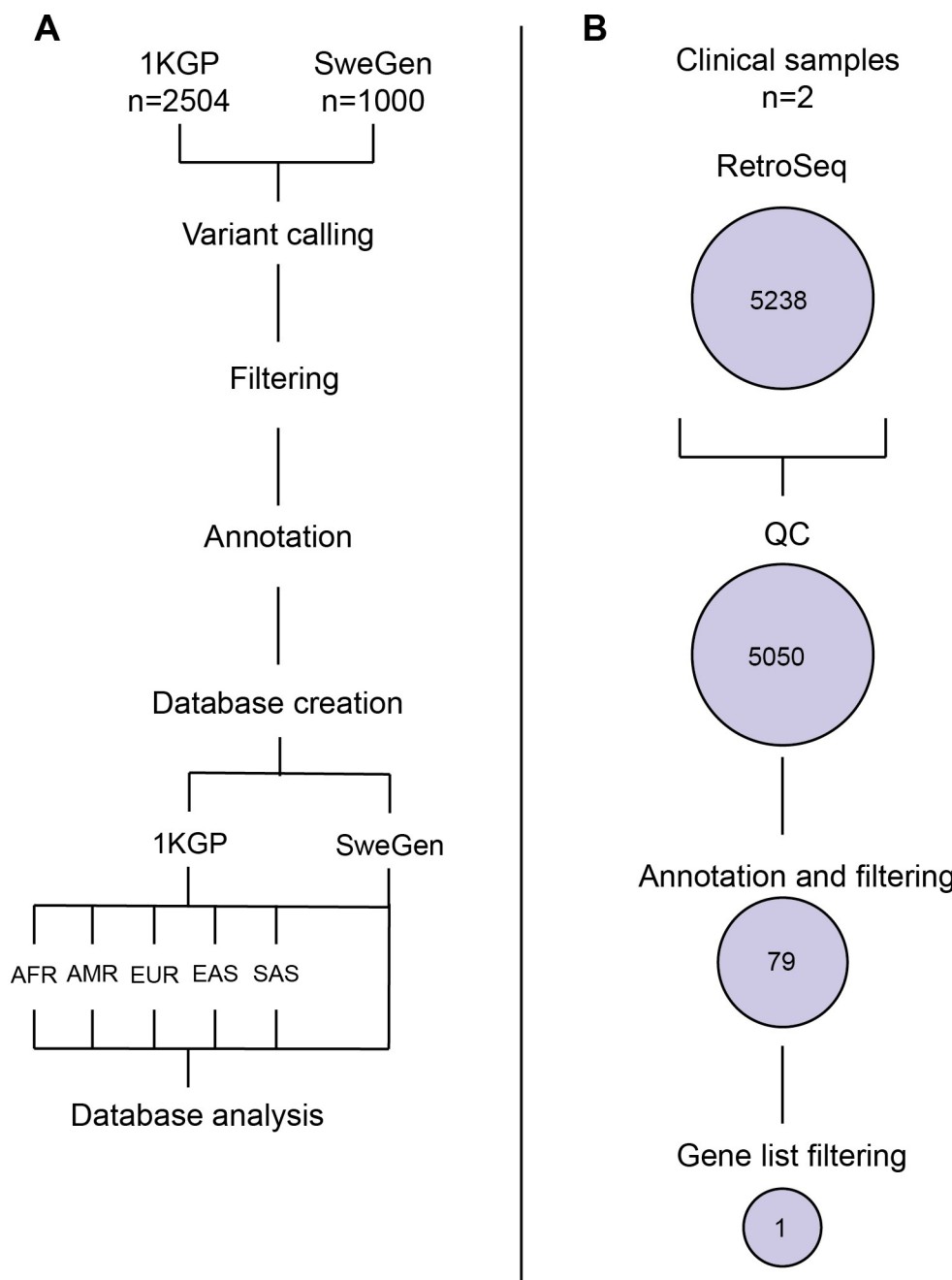

**Fig 1. Workflow overview for database generation and clinical analysis. (A)** workflow for generating populational TE insertion databases, and **(B)** number of calls in the clinical samples following the major steps of the workflow.

The TE insertions were annotated using Variant Effect Predictor (VEP). TE insertion affecting protein coding genes (n = 18704); with a median of 67 protein coding genes affected by TE insertions per individual. These TE insertions were matched with gene lists for intellectual disability (n = 2391), cerebral malformations (n = 107) and osteogenesis imperfecta (n = 185) (Fig 5).

TE insertions are abundant in protein coding genes, with a median of ~0.8 TE insertions per gene across *Alu*, L1, SVA and HERV. When applying the intellectual disability gene list,

**Table 1. Per sample mean, standard deviation (SD) and median for TE insertions identified >1% of individuals across the 1KGP and SweGen dataset.**

| TE | Mean | | SD | | Median | |
|---|---|---|---|---|---|---|
| | **1KGP** | **SweGen** | **1KGP** | **SweGen** | **1KGP** | **SweGen** |
| *Alu* | 4727 | 4213 | 2044 | 2490 | 4202 | 3460 |
| HERV | 110 | 205 | 18 | 55 | 108 | 193 |
| L1 | 1731 | 1660 | 404 | 1385 | 1661 | 1127 |
| SVA | 386 | 206 | 54 | 61 | 379 | 195 |

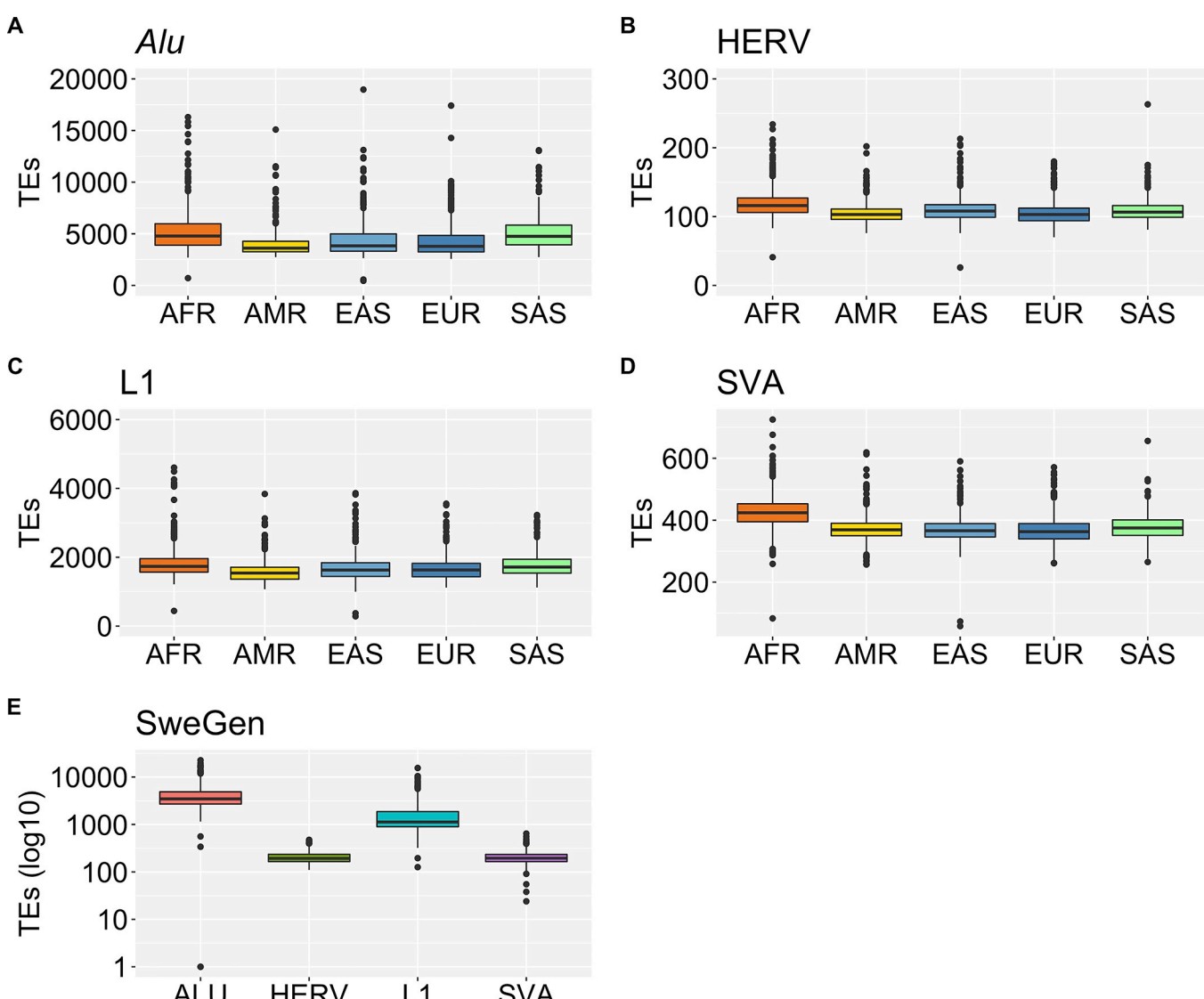

**Fig 2. Detected TE insertions (frequency (FRQ) > 1%) across the 1KGP and SweGen dataset. (A)** Amount of *Alu* insertions detected in the different populations of the 1KGP dataset. AFR African, AMR American, EAS East-Asia, EUR European and SAS South Asia. **(B)** Amount of HERV insertions detected across the 1KGP. **(C)** Amount of L1 insertions detected across the 1KGP dataset. **(D)** Amount of SVA insertions detected across the 1KGP dataset. **(E)** Amount of TE insertions *Alu*, HERV, L1 and SVA detected in the SweGen dataset.

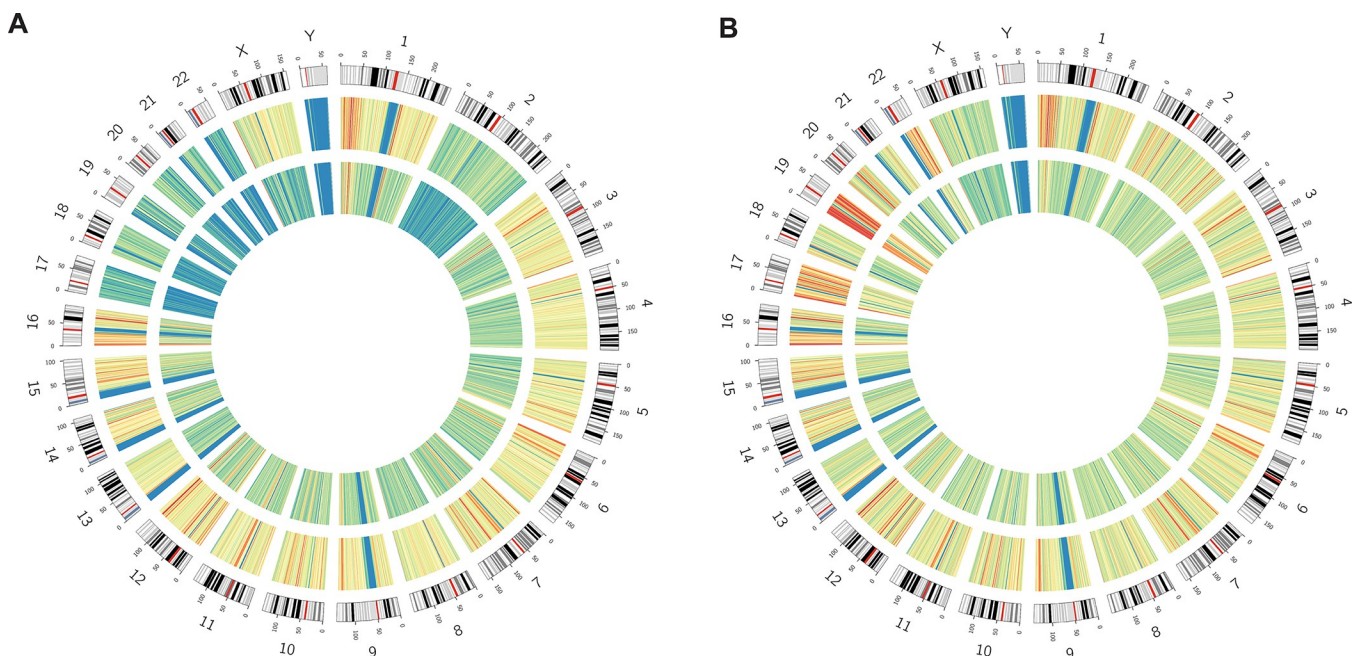

**Fig 3. Distribution of common TE insertions across the SweGen dataset.** (A) A circos plot depicting log base ratio of TE insertions present in >1% of individuals (outer track), and TE insertions present at >5% (inner track) across the SweGen database. (B) A circos plot depicting log base ratio of TE insertions present in >1% of individuals (outer track), and TE insertions present at >5% (inner track) across the 1KGP dataset. The blue and green colors indicate lower amounts of TE insertions per bin, whilst yellow and red indicate higher amounts.

only HERV affected genes had a median of 0 TE insertions per gene. *Alu*, L1 and SVA all have a median of ~0.1 TE insertion per gene in the intellectual disability list. As for TE insertions affecting genes in cerebral malformations or osteogenesis imperfecta lists, there was a median of 0 TE insertions per gene; except for L1 insertions in genes belonging to osteogenesis imperfecta, where the median was <0.05.

When applying allele frequency filtration as in Fig 4 (1KGP database frequency <1% (0.01)). Rare TE insertions were identified in 17126 protein coding genes, with a median of 151 per individual. No TE insertions were identified in genes from the cerebral malformations list. *Alu* and L1 had a median <0.05 in the intellectual disability list, and L1 had a median <0.05 in the osteogenesis imperfecta list (S1 File). The decrease in both total genes per TE and matches with gene lists proves the efficiency of filtering variants using gene lists in combination with a populational frequency filter.

## Clinical cases

Next, we applied the described TE insertion calling and filtering on two clinical samples (Fig 1). The two clinical cases originated from cohorts of patients with suspected neurofibromatosis and retinitis pigmentosa analyzed at the Karolinska clinical genetics diagnostic laboratory. Disease-causing TE insertions had previously been suspected in these cases, but not identified by our in-house SV pipeline [2].

We processed the already generated WGS data and detected a total of 5523 (2665 *Alu*, 104 HERV, 2161 L1, 419 SVA) and 4578 (2659 *Alu*, 102 HERV, 1385 L1, 318 SVA) TE insertions in patients A and B respectively, of which <1% overlapped with genes. Exonic and rare TE insertions were further filtered using gene lists for neurofibromatosis (n = 9) and hearing loss (n = 139) (Fig 1, S2 File) resulting in a single TE insertion per individual. Our workflow

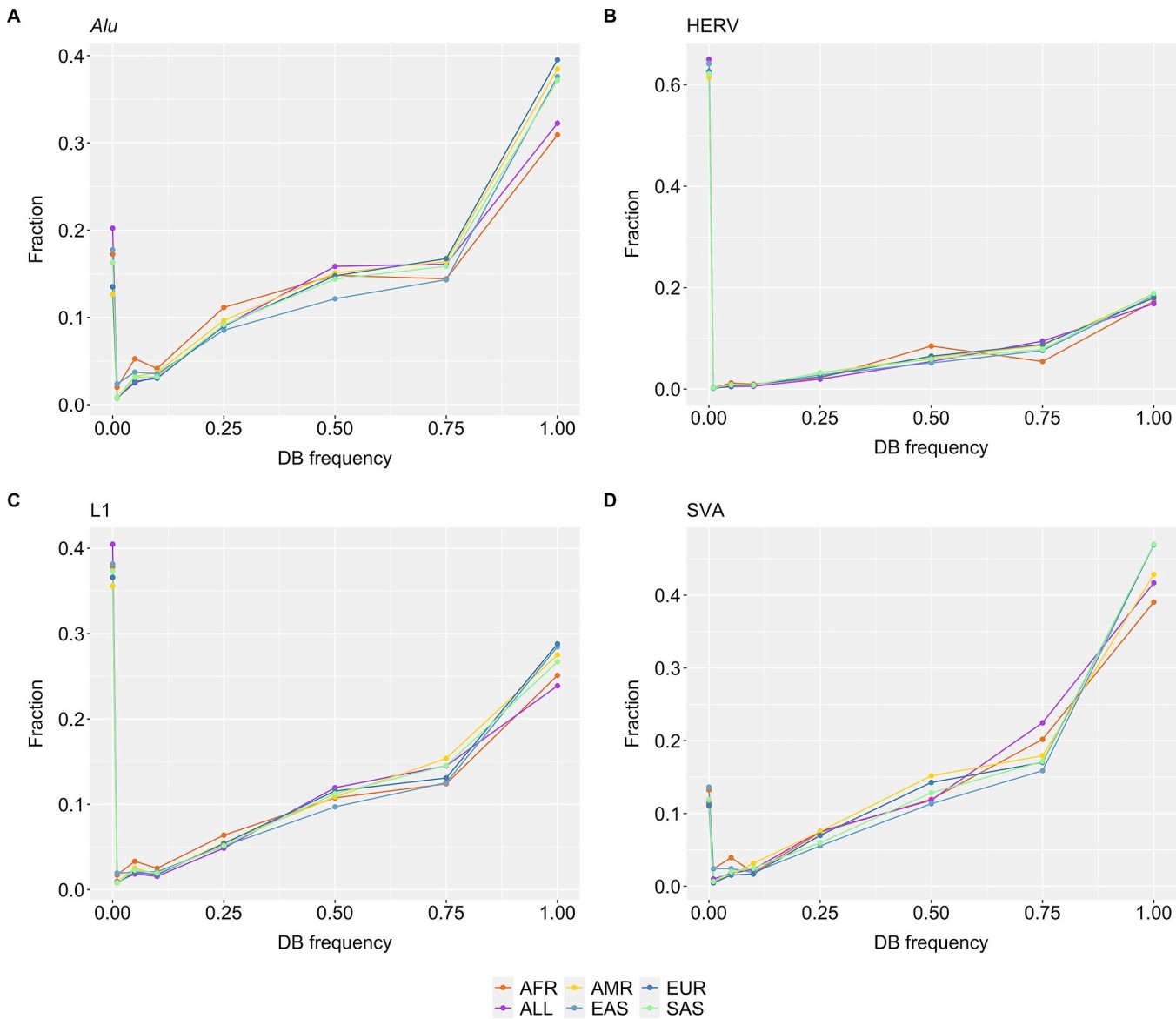

**Fig 4. Mean fraction of TE insertions identified in the SweGen dataset.** Fraction of SweGen TE insertions with a certain allele frequency in the databases 1KGP databases AFR, ALL populations, AMR, EAS, EUR or SAS. The specific TE subcategories are illustrated separately, **(A)** *Alu* **(B)** HERV **(C)** L1 and **(D)** SVA.

successfully identified the disease-causing *Alu* insertions in exon 37 of *NF1* and exon 45 of *USH2A*, both TE insertions heterozygous (Fig 6, S1 File). The exonic *Alu* elements lead to splicing defects and loss of functional protein by interrupting the normal exon (Fig 6B and 6D). The heterozygous *Alu* insertions in *NF1* was the cause of autosomal dominant Neurofi-bromatosis, type 1 (MIM 162200) in patient A. In the case of *USH2A*, the *Alu* insertions was in trans with a known pathogenic splice variant (c.5573-834A>G), causing autosomal recessive Usher syndrome, type 2A (MIM 276901).

To estimate the abundance of disease-causing TE insertions, we checked the results of all genetic testing done on these cohorts. Altogether, 80 cases with suspected neurofibromatosis had been tested, and in 27% (n = 22) the pathogenic or likely pathogenic variant was identified

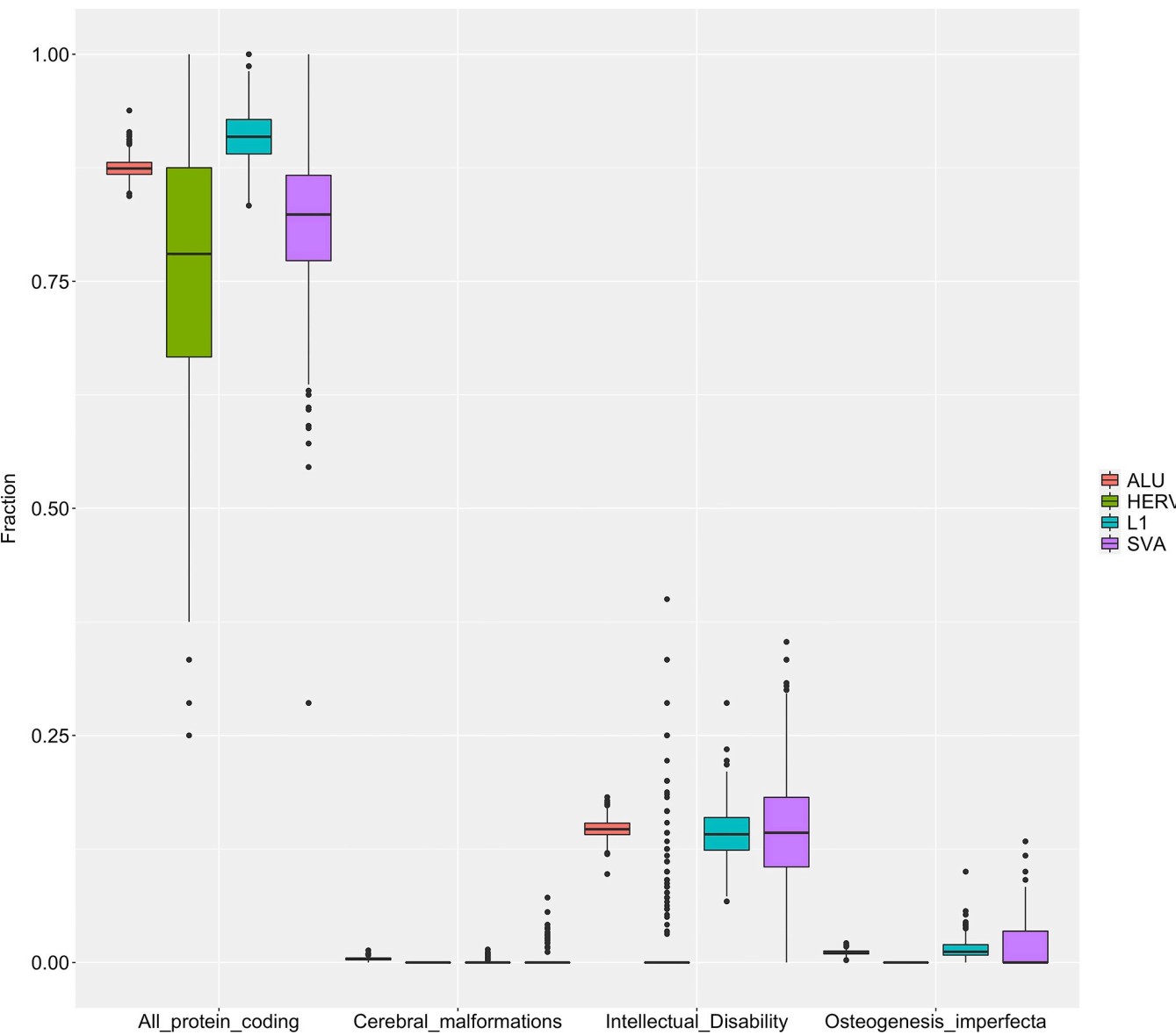

**Fig 5. TE insertions overlapping with genes in gene lists.** Fraction (TE insertions in protein coding gene from list/all protein coding TE insertions–per TE type) of *Alu*, HERV, L1 and SVA elements in the SweGen present in the 1KGP dataset, affecting protein coding genes present in; all protein coding (n = 22882), cerebral malformations (n = 107), intellectual disability (n = 2391), or osteogenesis perfecta (n = 185).

(S2 File). In two of these cases an *Alu* insertion was the pathogenic variant, making the fraction of disease-causing TE insertions 2.5% (2/80) in this cohort. Similarly, 54 patients with retinitis pigmentosa had been tested, where the pathogenic or likely pathogenic variants were detected in 51% (n = 28) of cases, where one (1.8%, 1/54) was an *Alu* insertion.

## Discussion

In this study we applied a TE insertions identification and annotation workflow and created a public database of common insertions across 1000 Swedish- and 2504 mixed-population individuals which can be used in clinical analysis. Furthermore, we show that TE insertions are

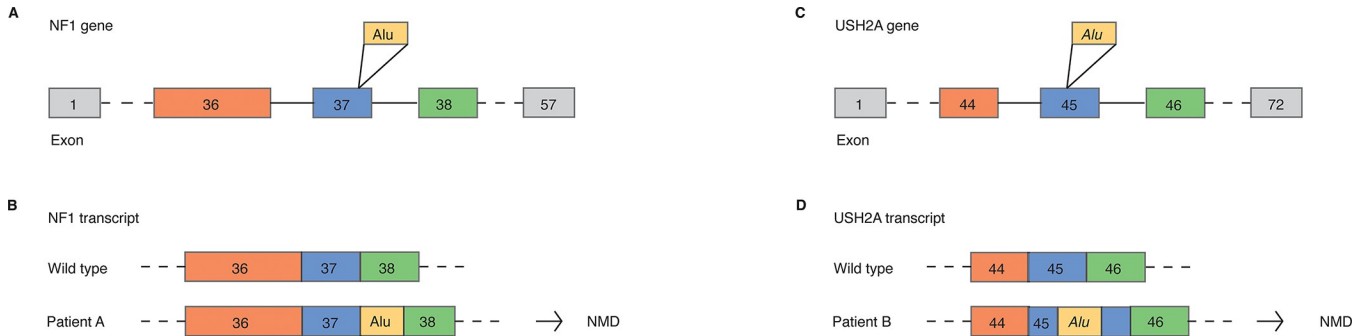

**Fig 6. Disease-causing *Alu* insertions identified using the pipeline. (A)** Overview of *Alu* insertion in the *NF1* gene. **(B)** Result of *Alu* insertion in *NF1*, causing disease phenotype. **(C)** Overview of *Alu* insertion in the *USH2A* gene. **(D)** Result of *Alu* insertion in *USH2A* exon 45, causing disease phenotype.

common across populations and that our workflow can efficiently identify disease-causing TEs.

Although TE insertions constitute only a small fraction of the known disease-causing genetic variants [17], it has been shown that both common and rare insertion events may impact gene expression levels [28] indicating that TE insertions may contribute to phenotypic diversity and disease. We detected higher amounts of TE insertions than the previous TE focused studies. In total, SweGen harbored 28% more TE insertions than 1KGP (135 719 and 105 742 respectively), much higher than the thousand genomes SV map (16 631) [6] and the GnomAD SV release (77 586) [9]. These discrepancies are likely explained by differences in the TE insertion detection methodology applied. In particular, the thousand genomes SV map and GnomAD SV release uses MELT [29] for TE insertion detection. Here, we utilize the open access specialized TE caller RetroSeq [26], which has a light computational footprint, overall well performance and high reliability, as shown in benchmarking studies [30, 31]. Studies report a low false discovery rate of 7.7% and a false negative rate at 6.7%, with a sensitivity close to 75% and precision >95% [26, 30, 31]. Regardless, in the clinic as well as in research, a more correct annotation of SV calls from WGS data is desirable.

Herein, we use RetroSeq, calling TE insertions based on split reads and discordant read pairs [26]; however, since the TE insertions are clearly recognizable in IGV [32] (S1 File), a more mature algorithm could benefit from image processing combined with machine-learning. A similar approach has been pioneered by the Deepvariant tool [33] and SVision [34] utilizing image processing to detect SNVs, INDELs and complex SVs. Although a large number of TE insertions were detected, it is clear that the current WGS TE insertion callers perform poorly [31, 35, 36] indicating the need of large scale long read population genomic studies to fully understand the frequency and impact of TE insertions. One major limitation of this study is that only the two disease causing TEs, identified in clinical samples, were validated with PCR. Unfortunately, we did not have DNA samples from the 1KGP and SweGen. However, RetroSeq is well tested [26] and has been shown to offer high sensitivity and precision as discussed above. Altogether this indicate that the calls are indeed reliable. Future studies using long read sequencing will likely be even better at identifying and genotyping repeat elements [36, 37].

Regions with high gene density, GC content and sequence diversity, such as chromosomes 1, 6 and 19, were more TE-dense (Fig 3). A considerable number of TE insertions across the SweGen dataset were genic (n = 4000, 3% of all TE insertions identified), 70% of these were found in genes present in the "all protein coding genes" list, indicating that TE insertions may play a role in both normal traits and disease. However, it must be noted that the ENSMBL

gene annotation [38] is extensive [39], and in total the protein coding genes cover roughly 40% of the human genome, partly explaining the large number of observed genic TE insertions. These genic TE insertions were not found at a high rate in genes from disease-specific gene lists, highlighting the efficiency of filtering using gene lists.

TE insertions can result in gene disruption, frameshift mutations and altered splicing [9] but routine TE insertion calling is generally not performed in clinical genome analysis of rare disease patients. In consequence, disease-causing TE insertions may be missed. Some variants might be detected by broad SV callers, but high confident detection of TE insertions require specialized callers as is demonstrated in the current article by the varying number of insertions identified across studies [6, 9]. Previous studies in exome sequencing data has indicated TE insertions as the causative variant in 0.03–0.04% of cases; we believe this number to be under-estimated, due to the difficulty of pinpointing TE insertions in such datasets [17, 40]. As a con-sequence, we believe TE insertion detection could increase the diagnostics yield in the clinic, highlighting their diagnostic value.

In the clinic, it is important to minimize the variants of unknown significance and since most TE insertions are shared across populations, we extracted the TE insertions across the SweGen data for further clinical interpretation (Fig 3). Specifically, by using the 1KGP TE insertion database as frequency filter, all variants present at frequencies higher than 1% could be removed in the SweGen cohort, removing >66% of all calls. Similar results were observed when using the SweGen database on the two clinical samples. By filtering out variants present at >1% in the databases, we generate a short variant list of 79 that can be manageable, or with gene list filtration, be manageable in the clinic (Fig 1). This demonstrate the importance of having populational databases that capture the common variants in a population.

Further enhancing the efficiency of gene lists, TE insertions in the SweGen data was filtered using gene lists. We found common protein coding genes affected by TE insertions to be pres-ent in the intellectual disability gene list (Fig 5). These findings could be an artifact of a com-prehensive gene list, but additionally indicate that common TE insertions may be of importance in neurodevelopmental disease. Unfortunately, we don't have access to appropri-ate tissue for validation experiments and our findings need to be validated further by analyzing expression levels of those genes in both TE insertion carriers and non-carriers. Regardless, our findings are supported by studies describing L1 transposition during neurogenesis [41], as well as in the hippocampus [10, 42, 43].

Illustrating the usability of TE insertion calling, we present two clinical examples where the disease-causing variant was pinpointed. In both the example disease-cohorts; neurofibromato-sis and retinitis pigmentosa, a large fraction of cases remain unsolved (73% and 49% respec-tively). There are several other examples where *NF1* has been the subject of disease-causing TE insertions [44, 45]. Studies reveal that TE insertions are responsible for 0.4% of all *NF1* muta-tions [45]. There are a vast amount of other diseases where TE insertions are involved, such as in cancers, autoimmunity and neuropsychiatric diseases [10]. Implementing systematic calling of TE insertions could identify additional cases across all rare genetic diseases. With many cases remaining undiagnosed, TE insertion calling in clinical genomes will likely contribute to an increased diagnostic yield.

## Methods

### Study subjects and datasets

Two publicly available datasets were used, SweGen (n = 1000) [24] and the 1000 genomes proj-ect (1KGP, n = 2504) [25]. In both WGS datasets, samples were sequenced to an average of 30x coverage and aligned to reference genome GRCh38. SweGen consists of 1000 Swedish

individuals representing the genetic variation in the Swedish population. In brief, the individuals were selected from the Swedish Twin Registry, a nationwide cohort of 10 000 Swedish-born individuals [24]. The 1KGP provides a resource on human genetic variation across 26 populations and 5 regions, including 661 African (AFR), 347 American (AMR), 504 East Asia (EAS), 503 European (EUR) and 489 South Asian (SAS) individuals [25]. The samples in each dataset were subsequently checked using samtools flagstat, where samples with a mapping quality, percent duplications or percent mapped ±2*SD were excluded from further analysis, which was 98 samples in SweGen and 146 samples in 1KGP.

At the Karolinska clinical genetics diagnostic laboratory (Stockholm, Sweden) WGS and *in silico* gene panel analysis is routinely used to diagnose patients with various clinical symptoms of rare diseases, with > 10 000 cases processed to date [2]. In two cases (Patient A and Patient B), pathogenic TE insertions in *NF1* and *USH2A*, had been identified prior to this study. For both, WGS was performed as described previously [2]. The clinical symptoms included signs of neurofibromatosis type 1 for Patient A and hearing loss for Patient B. Ethics approval for analysis of patient samples was given by the Regional Ethical Review Board in Stockholm, Sweden (ethics permit numbers 2012/222-31/3). This ethics permit allows for use of clinical samples for analysis of scientific importance as part of clinical development. Our IRB approval does not require us to get written consent for clinical testing. The research conformed to the principles of the Helsinki Declaration.

## Detection of transposable elements

The detailed workflow for TE insertion detection and filtering in the study is shown in Fig 1. In brief we used RetroSeq v1.5 [26] for TE insertion calling.

RetroSeq calls TE insertions using a two-step process. In the discovery stage, discordant read pairs are assigned a TE group (*Alu*, HERV, L1, SVA) using files with TE locations in the reference genome (-refTE parameter) [26]. In the following calling stage, output from the discovery phase is clustered and breakpoints are checked to determine TE insertion. The tool is one that can be easily implemented into clinical use and has been proven efficient with good sensitivity in previous studies [30, 31].

## Database of common transposable elements

After variant calling, TE insertion calls were compiled in a database using an in house developed software SVDB v 2.4.0 [46]. Seven databases were built, one for the SweGen dataset and the full 1KGP dataset, as well as one for each 1KGP subpopulation.

## Annotation and filtering

The genomic position of TE insertions were annotated using Variant Effect Predictor 99 [38] and tabix [47] for exon annotation [48]. TE insertion calls from SweGen [26], Patient A and Patient B were annotated with the population frequency (FRQ) of the TE insertion in the generated databases. Phenotype specific gene lists were obtained from the genomics England PanelApp [49] for neurofibromatosis (n = 9) and hearing loss (n = 139) (S2 File).

## Characterization of transposable element insertions in protein coding genes

Next, to assess whether TE insertions were affecting common genes, we extracted all TE insertions from the SweGen data that were present in the 1KGP databases which affected genes. The genes were compared with genes listed for intellectual disability (2391 genes), Cerebral

malformations (107 genes) and osteogenesis imperfecta (185 genes) downloaded from Pane-lApp [49], as well as a list of all protein coding genes (n = 22882) downloaded from Ensemble BioMart (S2 File).

## Supporting information

**S1 File.**
(DOCX)

**S2 File.**
(XLSX)

## Acknowledgments

The computations were performed on resources provided by SNIC through Uppsala Multidisciplinary Center for Advanced Computational Science (UPPMAX) under Project SENS2017106. One author (AL) of this publication is a member of the European Reference Network on Rare Congenital Malformations and Rare Intellectual Disability ERN-ITHACA [EU Framework Partnership Agreement ID: 3HP-HP-FPA ERN-01-2016/739516]x.

## Author Contributions

**Conceptualization:** Daniel Nilsson, Jesper Eisfeldt, Anna Lindstrand.

**Data curation:** Adam Ameur.

**Formal analysis:** Kristine Bilgrav Saether.

**Funding acquisition:** Anna Lindstrand.

**Investigation:** Kristine Bilgrav Saether.

**Methodology:** Jesper Eisfeldt.

**Project administration:** Anna Lindstrand.

**Resources:** Håkan Thonberg, Emma Tham, Adam Ameur, Jesper Eisfeldt, Anna Lindstrand.

**Software:** Kristine Bilgrav Saether.

**Supervision:** Jesper Eisfeldt, Anna Lindstrand.

**Validation:** Håkan Thonberg, Emma Tham.

**Visualization:** Kristine Bilgrav Saether.

**Writing – original draft:** Kristine Bilgrav Saether, Daniel Nilsson, Håkan Thonberg, Emma Tham, Adam Ameur, Jesper Eisfeldt, Anna Lindstrand.

**Writing – review & editing:** Kristine Bilgrav Saether, Daniel Nilsson, Håkan Thonberg, Emma Tham, Adam Ameur, Jesper Eisfeldt, Anna Lindstrand.

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
