## [Decision Letter · Decision Letter 0]

5 Apr 2023

PONE-D-23-00875Transposable element insertions in 1000 Swedish individualsPLOS ONE

Dear Dr. Eisfeldt,

Thank you for submitting your manuscript to PLOS ONE. After careful consideration, we feel that it has merit but does not fully meet PLOS ONE’s publication criteria as it currently stands. Therefore, we invite you to submit a revised version of the manuscript that addresses the points raised during the review process.

We look forward to receiving your revised manuscript.

Kind regards,

Osman El-Maarri, Ph.D

Academic Editor

PLOS ONE

Journal Requirements:

“AL recieved funding from, Stockholm Region (FoUI-961630, FoUI-954569), the Swedish Research Council (2017-02936, 2019-02079), and Karolinska Institutet.”

“The work was supported by the Swedish Research Council (2017-02936, 2019- 395 02079), Karolinska Institutet and the Stockholm Region (FoUI-961630, FoUI-954569). The 396 funders had no role in study design, data collection and analysis, decision to publish, or 397 preparation of the manuscript.”

“AL recieved funding from, Stockholm Region (FoUI-961630, FoUI-954569), the Swedish Research Council (2017-02936, 2019-02079), and Karolinska Institutet”

Reviewers' comments:

Reviewer's Responses to Questions

**Comments to the Author**

1. Is the manuscript technically sound, and do the data support the conclusions?

Reviewer #1: Yes

Reviewer #2: Partly

2. Has the statistical analysis been performed appropriately and rigorously? 

Reviewer #1: Yes

Reviewer #2: No

3. Have the authors made all data underlying the findings in their manuscript fully available?

Reviewer #1: Yes

Reviewer #2: Yes

4. Is the manuscript presented in an intelligible fashion and written in standard English?

Reviewer #1: Yes

Reviewer #2: Yes

5. Review Comments to the Author

Reviewer #1: The manuscript is focused on the transposable elements insertion events in 1000 swedish population. 66% of TE

insertions in SweGen were present at >1% in the 1KGP databases. Although ~0.7% of those insertions affect protein coding genes, they rarely affect known disease casing genes (<0.1%). Those data are clear and enough to publish to PLOS ONE.

Reviewer #2: Review

The manuscript was aimed to analyze transposable element insertions in 1000 Swedish individuals and they create a population specific TE insertion data base. They also tried to evaluate the TE insertion workflow on two clinical cases where pathogenic TE insertion were suspected. There are some interesting findings presented, however I believe the manuscript is not currently of sufficient quality to be considered suitable for publication. I have set out my reservations below.

Major Concerns:

I find that the analyzes are unreliable because we cannot rely on references to have insertions without resorting to statistical tests. It is better to use the Kmeans method. Furthermore, the RetroSeq tool used in this study is not specific to insertion and is based on orthology. Furthermore, given that the authors analyzed short reads, it is therefore essential to verify the insertions found by PCR or to validate them using another tool.

Minor Concerns :

1/ Keywords: mobile elements are synonym to transposable elements.

Keep one of the two keywords.

2/ Introduction :

Line 42-43: “The reasons for this could be due to the specific disease-causing gene not being described or that the underlying cause is not (mono)genetic”. It could also be due to epigenetic factors.

Line 55: The sentence « TE are DNA sequences that compose around 50% of the human genome [6].TE are DNA sequences that can change their position in the genome should be reformulated.” Should be replaced by« TE are DNA sequences that compose around 50% of the human genome that can change their position in the genome [6].”

Line 56-58: authors have to precise that class II elements transpose via DNA with cut paste mechanism (which may explain the 2% proportion) whereas class I transpose with copy paste mechanism.

6. PLOS authors have the option to publish the peer review history of their article (what does this mean?). If published, this will include your full peer review and any attached files.

Reviewer #1: No

Reviewer #2: No

---

## [Author Response · Author response to Decision Letter 0]

28 Apr 2023

Dear Dr El-Maarri

We would like to thank you for the much-valued comments and advises regarding our manuscript “Transposable element insertions in 1000 Swedish individuals”. We are pleased to hereby re-submit the article for your consideration. All reviewer comments have been carefully considered and addressed accordingly. Page numbers refer to the track changes version of the manuscript.

Reviewer #1

The manuscript is focused on the transposable elements insertion events in 1000 swedish population. 66% of TE insertions in SweGen were present at >1% in the 1KGP databases. Although ~0.7% of those insertions affect protein coding genes, they rarely affect known disease casing genes (<0.1%). Those data are clear and enough to publish to PLOS ONE.

Answer: Thank you for this nice evaluation of our article!

Reviewer #2

The manuscript was aimed to analyze transposable element insertions in 1000 Swedish individuals and they create a population specific TE insertion data base. They also tried to evaluate the TE insertion workflow on two clinical cases where pathogenic TE insertion were suspected. There are some interesting findings presented, however I believe the manuscript is not currently of sufficient quality to be considered suitable for publication. I have set out my reservations below. 

Major Concerns:

I find that the analyzes are unreliable because we cannot rely on references to have insertions without resorting to statistical tests. It is better to use the Kmeans method. Furthermore, the RetroSeq tool used in this study is not specific to insertion and is based on orthology. Furthermore, given that the authors analyzed short reads, it is therefore essential to verify the insertions found by PCR or to validate them using another tool.

Answer: The reviewer raises an important point, how do we know if the TEs identified in this study are true. We agree that Kmeans is a powerful method, however we opted to apply publicly available validated tools in this study, rather than developing our own methods. The caller RetroSeq is well tested, with over 200 citations, and uses a wide array of statistics internally [1-3] . The core of the study is to apply RetroSeq on two public datasets SweGen and 1KGP as well as clinical cases to discover non -reference TE insertions. The TEs identified in the clinical samples are validated using PCR; unfortunately we do not have the possibility to verify the SweGen and 1KGP findings, as we do not have access to DNA from these individuals.

We have added this to the discussion on page 13.

“One major limitation of this study is that only the two disease causing TEs, identified in clinical samples, were validated with PCR. Unfortunately, we did not have DNA samples from the 1KGP and SweGen. However, RetroSeq is well tested [1] and has been shown to offer high sensitivity and precision as discussed above. Altogether this indicate that the calls are indeed reliable. Future studies using long read sequencing will likely be even better at identifying and genotyping repeat elements [4, 5]”

Minor Concerns:

1/ Keywords: mobile elements are synonym to transposable elements.

Keep one of the two keywords.

Answer: Thank you for pointing this out, we have removed the keyword ”mobile elements”.

2/ Introduction : 

Line 42-43: “The reasons for this could be due to the specific disease-causing gene not being described or that the underlying cause is not (mono)genetic”. It could also be due to epigenetic factors.

Answer: We agree with the reviewer that epigenetic factors can cause disease. We have rewritten the sentence.

“However, equally important is a lack of knowledge regarding the human genome composition and structure [6], such as noncoding variants, as well as epigenetic modifications [7, 8].”

Line 55: The sentence « TE are DNA sequences that compose around 50% of the human genome [6].TE are DNA sequences that can change their position in the genome should be reformulated.” Should be replaced by« TE are DNA sequences that compose around 50% of the human genome that can change their position in the genome [6].”

Answer: Thank you for the correction. We have updated the sentence according to the reviewers suggestion. 

Line 56-58: authors have to precise that class II elements transpose via DNA with cut paste mechanism (which may explain the 2% proportion) whereas class I transpose with copy paste mechanism.

Answer: Thank you, we have updated the sentence to be more clear.

“Two main classes are described, DNA transposons (Class II TE), which move through a cut and paste mechanism, make up less than 2% of human genomes, and the more common retrotransposons (RTs, Class I TE) that move through RNA intermediates.”

References: 

1. Keane TM, Wong K, Adams DJ. RetroSeq: transposable element discovery from next-generation sequencing data. Bioinformatics. 2013;29(3):389-90. doi: 10.1093/bioinformatics/bts697.

2. Rishishwar L, Marino-Ramirez L, Jordan IK. Benchmarking computational tools for polymorphic transposable element detection. Brief Bioinform. 2017;18(6):908-18. doi: 10.1093/bib/bbw072. PubMed PMID: 27524380; PubMed Central PMCID: PMCPMC5808724.

3. Vendrell-Mir P, Barteri F, Merenciano M, Gonzalez J, Casacuberta JM, Castanera R. A benchmark of transposon insertion detection tools using real data. Mob DNA. 2019;10:53. Epub 20191230. doi: 10.1186/s13100-019-0197-9. PubMed PMID: 31892957; PubMed Central PMCID: PMCPMC6937713.

4. Ameur A, Che H, Martin M, Bunikis I, Dahlberg J, Hoijer I, et al. De Novo Assembly of Two Swedish Genomes Reveals Missing Segments from the Human GRCh38 Reference and Improves Variant Calling of Population-Scale Sequencing Data. Genes (Basel). 2018;9(10). Epub 20181009. doi: 10.3390/genes9100486. PubMed PMID: 30304863; PubMed Central PMCID: PMCPMC6210158.

5. Zhou W, Emery SB, Flasch DA, Wang Y, Kwan KY, Kidd JM, et al. Identification and characterization of occult human-specific LINE-1 insertions using long-read sequencing technology. Nucleic Acids Res. 2020;48(3):1146-63. doi: 10.1093/nar/gkz1173. PubMed PMID: 31853540; PubMed Central PMCID: PMCPMC7026601.

6. Alcazer V, Bonaventura P, Depil S. Human Endogenous Retroviruses (HERVs): Shaping the Innate Immune Response in Cancers. Cancers. 2020;12(3):610. doi: 10.3390/cancers12030610. PubMed PMID: 32155827.

7. Hammarsjo A, Pettersson M, Chitayat D, Handa A, Anderlid BM, Bartocci M, et al. High diagnostic yield in skeletal ciliopathies using massively parallel genome sequencing, structural variant screening and RNA analyses. J Hum Genet. 2021;66(10):995-1008. Epub 20210420. doi: 10.1038/s10038-021-00925-x. PubMed PMID: 33875766; PubMed Central PMCID: PMCPMC8472897.

8. Eisfeldt J, Rezayee F, Pettersson M, Lagerstedt K, Malmgren H, Falk A, et al. Multi-omics analysis reveals multiple mechanisms causing Prader-Willi like syndrome in a family with a X;15 translocation. Hum Mutat. 2022;43(11):1567-75. Epub 20220723. doi: 10.1002/humu.24440. PubMed PMID: 35842787; PubMed Central PMCID: PMCPMC9796698.

---

## [Decision Letter · Decision Letter 1]

18 Jul 2023

Transposable element insertions in 1000 Swedish individuals

PONE-D-23-00875R1

Dear Dr. Eisfeldt,

We’re pleased to inform you that your manuscript has been judged scientifically suitable for publication and will be formally accepted for publication once it meets all outstanding technical requirements.

Kind regards,

Ruslan Kalendar

Academic Editor

PLOS ONE

Reviewers' comments:

Reviewer's Responses to Questions

**Comments to the Author**

1. If the authors have adequately addressed your comments raised in a previous round of review and you feel that this manuscript is now acceptable for publication, you may indicate that here to bypass the “Comments to the Author” section, enter your conflict of interest statement in the “Confidential to Editor” section, and submit your "Accept" recommendation.

Reviewer #1: All comments have been addressed

Reviewer #2: (No Response)

2. Is the manuscript technically sound, and do the data support the conclusions?

Reviewer #1: Yes

Reviewer #2: Yes

3. Has the statistical analysis been performed appropriately and rigorously? 

Reviewer #1: Yes

Reviewer #2: N/A

4. Have the authors made all data underlying the findings in their manuscript fully available?

Reviewer #1: Yes

Reviewer #2: Yes

5. Is the manuscript presented in an intelligible fashion and written in standard English?

Reviewer #1: Yes

Reviewer #2: Yes

6. Review Comments to the Author

Reviewer #1: The manuscript of Transposable element insertions in 1000 Swedish individuals is important issue in relation to non-coding sequences. Authors are well revised it. Now, it is proper to publish to PLOS ONE.

Reviewer #2: Comments to the Author

The authors have addressed all of my areas of concern from their previous revision.

7. PLOS authors have the option to publish the peer review history of their article (what does this mean?). If published, this will include your full peer review and any attached files.

Reviewer #1: No

Reviewer #2: No

---

## [Editor Report · Acceptance letter]

21 Jul 2023

PONE-D-23-00875R1 

Transposable element insertions in 1000 Swedish individuals 

Dear Dr. Eisfeldt:

I'm pleased to inform you that your manuscript has been deemed suitable for publication in PLOS ONE. Congratulations! Your manuscript is now with our production department. 

Kind regards, 

on behalf of

Professor Ruslan Kalendar 

Academic Editor

PLOS ONE